# Trauma Informed Child Welfare Systems—A Rapid Evidence Review

**DOI:** 10.3390/ijerph16132365

**Published:** 2019-07-03

**Authors:** Lisa Bunting, Lorna Montgomery, Suzanne Mooney, Mandi MacDonald, Stephen Coulter, David Hayes, Gavin Davidson

**Affiliations:** School of Social Sciences, Education and Social Work, Queen’s University Belfast, 6 College Park, Belfast BT7 1LP, Northern Ireland, UK

**Keywords:** trauma informed care, child welfare, adverse childhood experiences, rapid evidence review

## Abstract

Trauma informed care (TIC) is a whole system organisational change process which emerged from the seminal Adverse Childhood Experiences (ACE) study, establishing a strong graded relationship between the number of childhood adversities experienced and a range of negative outcomes across multiple domains over the life course. To date, there has been no systematic review of organisation-wide implementation initiatives in the child welfare system. As part of a wider cross-system rapid evidence review of the trauma-informed implementation literature using systematic search, screening and review procedures, twenty-one papers reporting on trauma-informed implementation in the child welfare system at state/regional and organisational/agency levels were identified. This paper presents a narrative synthesis of the various implementation strategies and components used across child welfare initiatives, with associated evidence of effectiveness. Training was the TIC implementation component most frequently evaluated with all studies reporting positive impact on staff knowledge, skills and/or confidence. The development of trauma-informed screening processes, and evidence-based treatments/trauma focused services, where evaluated, all produced positive results. Whilst weaknesses in study design often limited generalisability, there was preliminary evidence for the efficacy of trauma-informed approaches in improving the mental and emotional well-being of children served by community-based child welfare services, as well as their potential for reducing caregiver stress and improving placement stability.

## 1. Introduction

### 1.1. Trauma Informed Care

Trauma informed care (TIC) is a whole system organisational change process which seeks to embed theoretically coherent models of practice across diverse settings and roles, including child welfare, family support, justice, mental health and education. It emerged from the findings of the seminal Adverse Childhood Experiences (ACE) study in the U.S. [1] with subsequent international and UK research establishing the same, strong graded relationship between the number of childhood adversities experienced (inclusive of physical, sexual and emotional abuse, neglect and household adversity), and a wide range of negative outcomes across multiple domains over the life course [2,3,4,5,6]. 

In recognising the impact of childhood adversity on child and adult outcomes, trauma-informed services strive to build trustworthy collaborative relationships with children and the important adults in their lives, as well as improve consistency and communication across linked organisations and sectors, with the aim of mitigating the impact of adversity by supporting and enhancing child and family capacity for resilience and recovery. TIC also seeks to change organisational practices and reduce the use of some practices, such as restraint or seclusion, which may inadvertently exacerbate the detrimental effects of severe adversity and constrain client/service user engagement. TIC advocates have developed a set of key assumptions and principles to help design responsive, holistic and effective systems of care [7]. These include paying attention to experience at all levels of the system, not only the service user/identified client, but also their caregivers (both families and professional caregivers), as well as practitioners, service managers and inter-agency interfaces. TIC is inclusive of adversity and trauma-specific interventions (such as dedicated services and interventions for substance misuse, domestic violence, sexual violence or post-traumatic symptoms) while also incorporating trauma principles into the overall organisational culture.

### 1.2. Trauma Informed Care and the Child Welfare System

The child welfare workforce interfaces with children and adults who have experienced adversity and trauma on an everyday basis. Indeed, it can be argued that no other child-serving system encounters a higher percentage of service users with trauma histories, whether it be in family support, child protection, foster, kinship or residential care. Experiences of maltreatment and neglect, parental mental ill health, domestic violence and substance misuse often co-occur [1], while removal from the family home and multiple placement moves can present additional stressors. Although professionals are often aware of the circumstances which precipitated contact with the child welfare system, they may be less aware of the trauma history of the children and their parents/caregivers, and may not always link this with current behavioural or emotional problems, or have access to appropriate resources to address these needs. In order to become trauma-informed, child welfare systems not only need effective trauma screening and assessment protocols, but also access to research-based trauma treatment services beyond generic mental health services [8]. A wider systems approach that recognises the important role foster parents, adoptive parents, professional caregivers and courts can play in facilitating post-trauma recovery, is also necessary. 

In an effort to develop more trauma-informed child welfare systems, various national initiatives, practice and training models have emerged. Of particular note is the work of the National Child Traumatic Stress Network (NCTSN) in the United States. Established by Congress in 2000, the NCTSN is a group of 70 treatment and research centres across the United States which has been instrumental in implementing trauma-informed child welfare initiatives in the USA and internationally. The development of trauma-informed practice in child welfare has also seen substantial federal funding with the Administration for Children and Families (ACF), a division of the United States Department of Health and Human Services (HHS), funding five-year demonstration grants in 2011 to develop and evaluate a range of strategies for improving care for children in the child welfare system suffering from exposure to trauma. Strategies included workforce development, trauma screening and referral, dissemination of trauma-focused evidence-based treatments (EBTs), and improved collaboration between child welfare and behavioural health services.

Given that TIC requires multiple level change, various guidance and frameworks for implementing trauma-informed approaches in organisations have also been developed. While definitions and terminology vary, Hanson and Lang [9] identified common elements of TIC implementation in child welfare settings (Table 1). However, they also noted the absence of research evaluating large-scale TIC efforts, and critically, whether they produced the improved child and family outcomes or future cost savings envisaged. More recent systematic reviews on TIC implementation initiatives in varied settings [10,11,12,13,14] have also highlighted a number of methodological problems within the TIC literature. These include a developing but still relatively limited focus on child and family outcomes; the preponderance of research designs using small samples and lacking a control group; short follow-up periods; high attrition rates; and the inability to disentangle the effects of individual implementation components from broader project outcomes. Despite these limitations, the reviews concluded that: there is preliminary support for the efficacy of trauma-informed models to increase positive outcomes for children in out-of-home-care [11]; that comprehensive, theoretically grounded, and developmentally-informed trauma-informed initiatives can help to reduce the use of seclusion and restraint, and staff and patient injury rates in youth inpatient psychiatric and residential treatment settings [10], and that TIC interventions involving a training component appear to improve staff knowledge, attitudes and behaviours [12].

However, to date, there has been no systematic review of child welfare system initiatives encompassing the broader range of child welfare services outside of residential or out-of-home care. As part of a wider cross-system rapid evidence review of the trauma-informed implementation literature using systematic search, screening and review procedures [15], this paper presents a narrative synthesis of the various implementation strategies and components of community-based TIC child welfare initiatives, together with associated evidence of effectiveness.

## 2. Materials and Methods

### 2.1. Review Question

What are the key components of approaches used within systems of care to create trauma-informed practice and what is the evidence of their effectiveness?

### 2.2. Search Strategy

A systematic search for relevant articles was conducted on the 22nd October 2018 using the databases: International Bibliography of the Social Sciences (IBSS); Science Citation Index Expanded (SCI-EXPANDED)—1970–present); PsycINFO (2002–present); Ovid MEDLINE (ALL 1946 to 31 August 2018); SCOPUS; and ERIC (see Appendix A for full search strategy).

### 2.3. Selection Criteria

#### 2.3.1. Trauma Informed Care

A broad search strategy was used to identify articles with the terms “trauma-inform*“, “trauma inform*”, “trauma-focus*”, “trauma focus*”, “trauma-base*” or “trauma base*“, in the title, abstract, keyword or subject headings (see Appendix A). 

#### 2.3.2. Study Population

Research studies focused on organisational-level implementation of trauma informed care involving single or multiple professional disciplines, e.g., child welfare, education, justice, adult social care, primary care and mental health. Articles which reported only on trauma-specific treatments, such as cognitive behavioural therapy (CBT) or different forms of psychotherapy, were excluded, as were those which reported on a specific service/intervention but did not include wider, organisational implementation components.

#### 2.3.3. Outcomes

To be eligible for inclusion, papers needed to clearly identify the trauma-informed components of the initiative being implemented and contain some evaluation component with associated data. No limits were placed on the type of outcomes measured and papers could include a range of outcomes such as the impact on service users, improvements to staff knowledge and practice, organisational change, policy development etc.

#### 2.3.4. Study Design

As well as including an evaluation component, articles which only reported on training evaluations also needed to include a pre- and post-test evaluation design to be eligible for inclusion. Literature reviews based on systematic methods were included while non-systematic reviews were excluded, as were studies using a single case design. Additionally, non-English language papers, papers published before 2009, and conference proceedings, dissertations and other papers not published in journals, were excluded. 

### 2.4. Screening 

The search identified 5527 potentially relevant articles. References were exported to an excel database and 3824 duplicates removed (see Figure 1). The title and abstract of the remaining 2243 articles were each screened by two members of the research team. Where there was disagreement, these were resolved by a third team member. In total, 2118 articles were excluded at screening primarily because they were discussion-based papers with no data presented, or because they focused on trauma-specific treatments such as CBT, psychotherapy etc. or a specific service/intervention which did not include wider, organisational implementation components.

The full text of the remaining 125 articles was then reviewed against the eligibility criteria. Forty-five articles were excluded at this point, primarily because they did not present evaluation data, they only evaluated training with a post-test or qualitative design, or they were not systematic reviews. There was no full-text availability for an additional five papers.

The remaining seventy-five papers selected for data extraction contained five systematic reviews which identified definitions and components of TIC relevant to: the juvenile justice system [13]; out-of-home-care [11]; youth inpatient psychiatric and residential treatment settings [10]; inpatient mental health settings [14]; and organisation wide trauma-informed initiatives involving a training component [12]. Twenty-eight of the seventy-five papers were individual studies already included within these systematic reviews. 

### 2.5. Data Extraction and Synthesis

Data extraction entailed extracting key study data (country, system of care, setting, TIC implementation components, evaluation methods, sample size and population, key findings and study limitations) and exporting to an MS Excel spreadsheet. For those papers cited in systematic reviews, the systematic review was the primary source for data extraction supplemented, where necessary, with data from the original article.

A narrative approach to synthesis was adopted. Narrative synthesis relies primarily on text to summarise and explain the findings from multiple studies [16] and is commonly used when varied measures and outcomes make meta-analysis unfeasible. It involves summarising individual studies, grouping them according to relevant characteristics such as intervention type, population, design, setting or outcomes, and identifying commonalities and differences within and between groups. Studies included in this review were first grouped according to service system (child welfare, education, health, justice) and then subdivided according to common settings and/or geographical coverage as well as whether they reported on service user outcomes, specific elements of the implementation process and/or results from implementation evaluations. Implementation process and evaluation data were further categorised according to an adapted version of Hanson and Lang’s [9] domains of trauma-informed implementation. Results were presented primarily in narrative form and categorised using this framework. Examples of the ways in which various projects have implemented different TIC domains were reported, together with summaries of the key findings from associated evaluations and any relevant methodological issues. Full details of each of the child welfare initiatives, settings, TIC implementation components, evaluation design, measures, findings and limitations can be found in Appendix A.

## 3. Results

Of the 75 relevant papers identified through searching the academic literature, 21 papers reported on evaluations of 17 community-based child welfare initiatives involving frontline social workers, family welfare staff and/or other professionals. All originated in the USA. Eight initiatives were large state-wide initiatives usually comprising multiple TIC implementation components and covering multiple professions and agencies, primarily child welfare caseworkers, clinical staff, foster care and adoption services, family preservation services and/or child welfare/treatment facilities. Nine were organisational or agency-level initiatives which generally targeted staff employed in specific agencies such as Child Advocacy Centres, fostering agencies or family preservation services. Across initiatives, service user and training outcomes were the most commonly evaluated elements of the TIC initiatives, followed by evidence-based treatment and trauma-focused services (see Table 2). Evaluation of service user, treatment and training outcomes were primarily empirical (see Table 3 and Table 4), while reference to other implementation components, were primarily descriptive.

### 3.1. Service User Outcomes

Only eight projects reported specifically on outcomes for children and/or their families (see Table 3). The state-wide Massachusetts Child Trauma Project (MCTP) was the largest and the most comprehensively evaluated of these, reporting that the 55,145 children who received MCTP interventions had significantly lower substantiated maltreatment reports when compared to the 36,108 children who had not received the intervention [17]. However, children in the intervention group had more maltreatment reports overall (both substantiated and unsubstantiated) and total out-of-home placements than did their counterparts in the comparison group. This finding was potentially related to increased surveillance and reporting of maltreatment and placement issues by MCTP’s trained child welfare caseworkers and treatment providers. Another state-wide initiative to embed evidence-based, trauma-informed practices into the child welfare and mental health systems [18], found that carers’ perceptions of trauma-informed services moderated the relationship between child behavioural health needs and carer satisfaction and commitment. This relationship was evident in relation to foster parents (but not adoptive parents) perceptions of trauma-informed mental health services (but not child welfare services) suggesting these services can act as a buffer and potentially improve placement stability for this group.

Six organisational-level initiatives also evaluated case outcomes. Two initiatives involving child protection/family preservation services reported reductions in child behaviour problems following implementation of the Attachment, Regulation and Competence (ARC) model in a community trauma treatment centre [19], and increased family safety, caregiver capabilities and child well-being [20] following participation in trauma-informed family preservation services. One community project for at-risk female youth in Hawaii [21] also found significant improvements from baseline to six-month follow-up on measures of youth strengths, competence, depression, impairment, behavioural problems, emotional problems, as well as caregiver strain. Similarly, two fostering/adoption services reported reduced child mental health and behavioural difficulties [22,23], as well as reduced care-giver stress [22] and improved placement stability [23]. A pilot of Intensive Permanence Services (IPS) [24] noted that 80% of the young people who were involved in the project and who completed at least 13 months, achieved legal permanency while youth who did not complete the IPS did not achieve legal permanency at the same rate (exact numbers not provided). Suarez et al.’s [21] evaluation also included a financial analysis which indicated that these outcomes were obtained with a minimal overall increase in costs when compared to standard care alone ($365,803 vs. $344,141).

With the exception of the MCTP outcome evaluation [17], most studies lacked a control or comparison group and were based on small sample sizes. As such, while positive trends were observed, the effectiveness of large scale, system wide initiatives remain an area requiring significant further evaluation. Despite the lack of a control group, the KVC Behavioural Healthcare evaluation [23] employed an innovative analytical approach, using the extent to which staff members had been trained in Trauma Systems Therapy (TST) and showed fidelity to the TST model at quarterly supervision sessions to calculate children’s exposure to TST “dosage”. While it might be expected that overall TST “dosage” amongst members of the care team who worked most closely with the children would be associated with significant improvements, more surprisingly, the “dosage” amongst those who worked more distally with the child was also significantly associated with improvements in functioning and placement stability, suggesting that it may be the understanding, skills and practice coherence of the child’s entire care team that produces better outcomes rather than specific individuals.

#### Training

Training was, by far, the most common component of TIC implementation and was described in almost all of the child welfare papers as a central element of the trauma-informed initiatives under review. Nine of the initiatives specifically evaluated training outcomes, mainly through quantitative pre-test/post-test designs (see Table 4), although a small number also used qualitative methods to identify agency progress [24,26,34], and one relied solely on qualitative interviews to assess implementation following initial training in the Sanctuary model [35]. Training content and delivery varied considerably in terms of duration, ranging from two hour training sessions to Child Health and Education Tracking (CHET) staff in Washington State [28], to involvement in year-long Learning Collaboratives for community mental health practitioners [25]. Training in state-wide initiatives generally targeted senior managers followed by front-line staff and were often based on training content developed by the National Child Traumatic Stress Network (NCTSN) with particular reference to in Child Welfare Training Toolkit, developed in conjunction the Chadwick Trauma-Informed System Project [25,26,29,30,31]. In order to establish a common understanding and language about trauma and its impact, Michigan Children’s Trauma Assessment Centre [32] employed one of the most comprehensive training strategies, developing and delivering training for specific groups, including schools, child welfare professionals, medical personnel, community mental health practitioners, foster carers and biological parents. 

Post-training follow-up ranged from six weeks to two years, with retention rates between 12% and 89%. Assessment was primarily based on self-reporting, although a number of studies utilised validated measures such as the Evidence-Based Practice Attitudes Scale (EBPAS), the Trauma-Informed System Change Instrument (TISCI) and the Trauma System Readiness Tool [25,26,31] to assess changes in practitioner attitudes and practice. However, regardless of follow-up timing or assessment measure, all studies reported increased staff knowledge, awareness and/or confidence in trauma-informed principles and practice that were retained over time. Although the absence of independent practice observation as a post-training measure represents a significant evidence limitation, findings indicated positive outcomes overall. Reported benefits included that practitioners were positive about evidence based practice and had strong intentions to consistently engage in trauma-informed practice [25], felt that their capacity to provide trauma-informed care had increased significantly as a result of training [31] and that their practice had become significantly more trauma-informed over time [26].

Similarly, participants in the “Training for Adoption Competency” initiative, which involved 855 professionals employed in mental health, adoption, family service and residential care agencies across 16 States [33], reported changes in their practice following the training at both individual and service levels of provision. “Training for Adoption Competency” was the only training evaluation to use a comparison group, showing substantial gains in TIC knowledge amongst the training group when compared to a group of similarly qualified professionals [33]. However, further details of the comparison group numbers, demographics and method of selection were not reported. In a qualitative evaluation of the “Lemonade for Life” initiative, home visitors and parent educators also reported that the TIC training materials provided tangible tools for use in work with families which increased engagement [34], although no supporting empirical evidence was provided.

### 3.2. Workforce Development

#### 3.2.1. On-Going Staff Support

Various initiatives stressed the importance of on-going staff support as crucial to maximising the impact of initial training and embedding TIC in practice. Strategies to address this included the use of learning collaboratives [25,31,32], coaching, mentoring and monitoring of fidelity to the trauma-informed model through supervision [27]; on-going consultation and coaching from model developments/trainers or other experts [32,33] and continuous staff training, booster sessions and/or recertification processes [18,27]. While there were no empirical evaluations of the effect these additional supports had on TIC implementation or staff and service user outcomes, qualitative evaluation of the KVC initiative [27] indicated that staff considered the multiple modes of training, and repeated exposure, as critical to successful implementation. KVC staff also reported valuing the additional supports that were provided e.g., professional role-specific workbooks, YouTube videos, email blasts to staff focused on specific TST topics, monthly staff and foster parent newsletters featuring articles on TST and “cheat sheets” (concise TST learning aids).

#### 3.2.2. Staff Support/Self-Care

Although staff care/self-care features as a key element of various TIC implementation frameworks, specific efforts to address this in community-based child welfare initiatives were limited with the exception of the Connecticut Collaborative on Effective Practices for Trauma (CONCEPT) [31] and the Michigan Children’s Trauma Assessment Centre [32]. The CONCEPT project created “Worker wellness” teams who provided quarterly trainings in self-care to a range of staff throughout the implementation process [31]. However, longitudinal assessment using the Trauma System Readiness Tool found that agency support for dealing with staff vicarious trauma was the lowest rated domain at both the start of the initiative and two years later. This would suggest that staff support is an area of TIC implementation that requires additional focus in child welfare settings. Michigan Children’s Trauma Assessment Centre identified trauma-informed child welfare decision-making as one of the greatest needs and most significant challenges within pilot communities and developed training to address issues of staff secondary traumatic stress and decision-making regarding removal of children from biological parents and placement changes. No specific evaluation data for this implementation element were reported, although overall improvements regarding the extent to which child welfare systems were trauma-informed were reported [32]. Additionally, a small-scale qualitative evaluation of TIC implementation strategies developed by leaders within social services agencies, also emphasised the importance of senior managers and supervisors being supportive of staff needs and showing concern for their well-being [35] but provided no further empirical evaluation. 

### 3.3. Trauma-Focused Services

#### 3.3.1. Screening and Assessment

A number of papers discussed the implementation of specific trauma screening processes, outcomes from training in routine inquiry, or described inclusion of evidence-based screening measures as part of trauma-informed training [26,29,32,34,37]. Qualitative evaluation of a routine inquiry initiative suggested that use of ACE screening increased engagement between home visitors and families [34]. However, only Lang et al. [37] and Henry et al. [32] presented evaluation data on the numbers of children screened. In a state-wide implementation of trauma screening in Massachusetts, Colorado, Connecticut, Montana and North Carolina [37], the target groups, screening tools and processes varied between states with some opting to screen children in all open cases, while others opted to screen children coming into care. Nonetheless, screening was generally perceived favourably by child welfare workers and mental health professionals. Not surprisingly, implementation led to significant increases in reported screening of children, although there remained wide variations. For example, in Massachusetts, the average rate of child screening increased from 40.3% to 75.0%, while in Colorado, 53% of open cases were screened over a 16-month period. 

In addition to presenting screening data, Lang et al. [37] noted that implementation of trauma screening in each of the five Child Welfare Systems had been a somewhat lengthy and challenging process in comparison with other implementation activities such as dissemination of evidence-based treatment and training staff in childhood trauma. While many of the challenges associated with trauma screening related to common systemic issues such as the size and scope of the Child Welfare System the number of staff, competing demands and staff turnover, the authors noted that the biggest barriers tended to be due to unique local issues such as IT systems, team cultures, limited buy-in and local availability of evidence based treatment.

In Michigan Children’s Trauma Assessment Centre [31], screening was carried out via the Trauma Screening Checklist (TSC). This was used to identify children and young people requiring targeted services as well as to leverage additional community resources. Over the course of implementation 964 screens were anonymously collected for children aged 0 to 17 years, primarily through child welfare workers, parents and school personnel. Results were used to raise awareness of the prevalence of children who had experienced childhood adversity which resulted in two project communities securing funding from local foundations for the development of trauma assessment centres. Approximately a third of Michigan’s local Community Mental Health agencies adopted the TSC for screening children at intake, with one agency completing 4500 screens. Implementation also entailed training three teams of professionals (out of the nine communities) in a comprehensive transdisciplinary neurodevelopmental assessment protocol suitable for children experiencing complex trauma. No evaluation data regarding the assessment process were however reported.

#### 3.3.2. Evidence-Based Treatment and Trauma-Focused Services

Four state-wide initiatives, the Massachusetts Child Trauma Project [25], the Arkansas state-wide initiative [29], CONCEPT in Connecticut [30] and the Michigan Children’s Trauma Assessment Centre [32], all incorporated strategies to build treatment capacity through training and dissemination of evidence-based treatments such as trauma-focused cognitive behavioural therapy (TF-CBT), child-parent psychotherapy and the Attachment, Regulation and Competency (ARC) model. For example, in the Arkansas project, trauma-informed training for child welfare staff was conducted following dissemination of training in TF-CBT to more than 150 mental health professionals across the state to maximise capacity for assessment and treatment referrals once child welfare workers were better informed about the effects of trauma on children. The Massachusetts Child Trauma Project [28] and Michigan Children’s Trauma Assessment Centre [33] presented data on the outcomes of this capacity building, noting that, over the course of the year of implementation, 298 and 230 children received treatment, respectively. Only the Massachusetts Child Trauma Project included further evaluation of the treatment provided. Findings demonstrated that, after approximately six months of treatment, children had fewer post traumatic symptoms and behaviour problems [26]. 

Other trauma-focused services provided as part of the implementation process included a 24 month, four phase programme to help youth in out-of-home placement achieve permanency and strengthen their connections to supportive adults [24]; structured group activities as well as evidence-based treatments such as TF-CBT and Girls Circle psycho-educational support groups [21]; the development of strengths-based, culturally appropriate, trauma-informed intake and family assessments accompanied by concentrated and family-focused case management services and referrals for material resources (e.g., housing, food, legal, transport, etc.) [20]; application of the ARC treatment framework as a brief outpatient intervention with adoptive children [22] and as a treatment model for families involved with child protection services [19]; school and community based trauma-informed interventions which included psycho-education, self-regulation skill building, trauma processing and safety planning (trigger management) [32]; and staff, birth parent and carer training and support in using Trauma Systems Therapy [23]. As noted in the service user outcomes section, service evaluation showed improvements in child mental and emotional well-being [19,20,21,22,23], decreased caregiver stress [21,22] and improved placement stability [22,24]. Likewise, although not linked to specific elements of service provision, participation in the Massachusetts Child Trauma Project [17] resulted in significantly lower substantiated maltreatment reports, while contact with trauma-informed mental health services improved foster parents’ satisfaction and commitment [18].

### 3.4. Organisational Change

#### 3.4.1. Leadership Buy-In and Strategic Planning

Many of the TIC initiatives reported were part of broader, organisation-wide trauma-informed implementation strategies aimed at changing organisational culture and practices. Key elements of implementation therefore focused on targeting leadership buy-in. This was achieved via providing initial training to agency directors and senior management, establishing implementation teams, developing strategic implementation plans and structures, and assessing organisation readiness [25,29,31,32,35]. Projects like the Michigan Children’s Trauma Assessment Centre emphasised more “grassroots” approaches centred on developing community partnerships and implementation strategies based on extensive collaborative community assessments and consultation [32]. This entailed using either the Trauma-Informed System Change Instrument or the Trauma System Readiness Tool to provide a baseline assessment of TIC policy and practice which, together with consultation with a wide range of key stakeholders, formed the basis for TIC implementation plans.

The Michigan Children’s Trauma Assessment Centre [32] placed a strong emphasis on local champions who mobilized resources and brought together community stakeholders who were perceived as central to system change. Their efforts were simultaneously supported by Michigan Children’s Trauma Assessment Centre’s participation in leadership meetings, reinforcing interest and momentum for change. As part of its implementation strategy, the Massachusetts Child Trauma Project [26] created Trauma-Informed Leadership Teams (TILTs) which focused on embedding and supporting a structure for TIC systems integration at the community level. These teams were led by a child welfare manager and supported by participation from social workers, service users, mental health providers and other community service providers/stakeholders. Qualitative evaluation identified these collaborative processes as central to the success of the both these projects [26,32]. 

#### 3.4.2. Developing Policy, Procedures and Data Systems 

A number of papers drew attention to changes made to policies, processes and/or data systems as part of the implementation process [26,27,30,31,32,37]. Two initiatives, the Massachusetts Child Trauma Project and the KVC Behavioural Healthcare initiative, presented empirical evaluation data, using the Trauma-Informed System Change Instrument [26,30] to measure self-reported changes to agency policy [26,32]. Both showed significant increases in staff perception that policy had become more trauma-informed one year after implementation. The KVC Behavioural Healthcare initiative [26] also emphasised the importance of using data to inform and monitor implementation progress. KVC enhanced its centralized data system to better monitor implementation, inputting information on staff and foster parent attendance at training as well as staff scores on fidelity assessments. Qualitative interviews with staff leadership and observations of leadership team calls indicated that these data were reviewed and discussed regularly, leading to the further development of implementation plans to promote continuous improvement [26]. 

In an effort to embed trauma-informed principles into decision-making processes, the Michigan Children’s Trauma Assessment Centre [32] developed a trauma-informed Court Report Checklist (CRC) to assist Family Court judges to understand a child’s trauma history, the impact of the trauma on their functioning and the services being provided the child. A file review conducted as part of the project evaluation, found that, prior to implementation, only three out of 53 Family Court files mentioned trauma and/or associated the child’s behaviour and emotional concerns with the impact of trauma. In contrast, two years following implementation of the CRC in the first pilot community, 100% of the cases had a CRC submitted by the child welfare worker to the judge prior to court hearings. The Michigan Children’s’ Trauma Assessment Centre also developed the Trauma-Informed Therapist Report as a method for therapists to inform caseworkers and the Family Court of progress in trauma-informed assessment and treatment.

Although not specifically evaluated, a number of other initiatives described efforts to develop trauma-informed policy and procedures. In the CONCEPT initiative in Connecticut [31], implementation entailed creation of a multidisciplinary core team supported by various subcommittees, including a policy workgroup. This group developed a standardised policy review tool to modify policies and practice guides to ensure consistency with trauma-informed principles. For example, the Family Assessment and Response (differential response) practice guide was modified to include consideration of the child’s and caregiver’s exposure to trauma, through assessing the common signs of traumatic stress in children and assessing the impact of the parent’s own trauma history on his or her ability to care for the child. At publication, nine policies and ten accompanying practice guides had been formally approved and disseminated to staff. Implementation of the NCTSN-adapted Child Welfare Referral Tool as a trauma screen for child welfare intake and assessment services in Massachusetts also led to the tool later being incorporated into existing assessment procedures [37].

#### 3.4.3. Service User Involvement and Changes to the Physical Environment

Although a number of initiatives reported on steps taken to engage service users as an integral component of implementing TIC in child welfare settings, the data included were largely descriptive with parents and carers the primary targets, rather than children and young people. Initiatives took the form of parent and carer involvement in trauma-informed training [25,27,32] and community engagement efforts [32]; service user involvement in leadership teams [25]; and engaging family members and other supportive adults as part of permanence planning for young people in foster care [38]. The grassroots approaches adopted by Michigan Children’s Trauma Assessment Centre [32] placed a strong emphasis on developing community partnerships and implementation strategies based on extensive collaborative community assessments which included foster parents and birth parents. This entailed using the Trauma-Informed System Change Instrument to provide a baseline assessment of TIC policy and practice which, together with key stakeholder consultation, formed the basis for TIC implementation plans.

The KVC Behavioural Healthcare initiative’s implementation of Trauma Systems Therapy (TST) provided one of the most comprehensive examples of a systems-wide training which initially targeted staff and foster parents and later expanded to encompass community partners and birth parents. It was also the only project to present process evaluation data which indicated that, after two years, 67% of foster parents had received trauma-informed training [26].

None of the community-based child welfare projects reported on changes made to offices or other facilities as a means of promoting a positive, safe physical environment in which to engage service users. 

## 4. Discussion

Previous reviews of the TIC implementation literature have highlighted a variety of methodological difficulties. These include a relative dearth of outcome focused evaluations, particularly in relation to large-scale TIC efforts, a lack of experimental designs, small sample sizes, high attrition rates and short follow-up periods in longitudinal designs [10,11,12]. In reviewing the child welfare TIC implementation literature, it is clear that while many of these difficulties remain, there has been a drive in recent years to develop the TIC outcomes evidence base with multi-component evaluations of large scale child welfare initiatives beginning to emerge [17,18,23,25,27]. These multi-method evaluations have drawn on administrative data, clinical assessments, staff surveys and interviews and focus groups with managers, staff, community stakeholders and service users in an effort to elucidate the impact of different TIC components. However, the difficulty in disentangling the effects of the various implementation components from the broader project outcomes persists. This is demonstrated in the disconnect between the range of implementation domains described in the child welfare literature and the much more limited extent to which empirical evaluation data were available for each of these. Even in the multi-method evaluations, it was generally not possible to isolate which implementation components contributed to implementation success. This reflects the challenges of evaluating whole organisation or system changes which are not well captured by standard evaluation methodologies which are designed to measure individual level changes [39,40,41]. This presents an ongoing challenge to evidence the inter-connected multi-level processes essential to effective whole system change envisaged by TIC.

The review methodology described in this paper also presents its own limitations when interpreting the findings from the child welfare TIC literature. Firstly, it was limited to organisational interventions that were explicitly trauma-informed. Although this allowed for the application of systematic and replicable methods to be applied to evaluate the body of evidence, it excluded interventions that embraced TIC principles without using the language of the trauma-informed. Secondly, as the review was concerned with effective approaches used within systems, it excluded specific trauma-informed clinical interventions and trauma-focused services/interventions which were not delivered as part of a wider programme of organisational change. Thirdly, although systematic search and data extraction methods were applied, this was a rapid evidence review rather than a systematic review and, as such, evaluation of research quality was limited to broad assessment of the study design and reported limitations. 

Despite these limitations, this review presents preliminary evidence for the efficacy of trauma-informed approaches in improving the mental and emotional well-being of children served by community-based child welfare services, as well as their potential for reducing caregiver stress and improving placement stability. Implementation at the workforce level focused primarily on staff training with all evaluations showing significant increases in staff knowledge, confidence and/or skills in applying TIC principles. These changes were maintained over time, in some cases up to one to two years after initial training. The development of specific TIC standardised measures such as the Trauma-Informed System Change Instrument (TISCI) and the Trauma System Readiness Tool (TSRT) brought additional rigour to these evaluations, enabling researchers/developers to assess baseline levels of knowledge of TIC policy and practice, assess system readiness, identify training and support needs and measure changes over time [25,26,31,32]. Although the TISCI and TSRT are both based on self-report and were used as a follow-up measure in only a small number of initiatives [26,32], they demonstrated significant increases in individual staff TIC knowledge and practices, as well as the extent to which organisational/agency policy and procedures had become more trauma-informed. 

Many initiatives also introduced strategies to provide on-going support to staff after initial training through supervision, booster training, coaching and mentoring. While the extent to which these contributed to overall outcomes was unclear, this implementation element was consistently emphasised as central to embedding TIC principles into every-day organisational practice. However, a focus on staff support/self-care was noticeably absent in this review with only two projects detailing specific efforts to address this [30]. To date, the limited number of studies investigating the impact of TIC initiatives on staff experience of trauma and stress in health and residential settings have produced mixed results. While some demonstrated no effect on staff turnover, job satisfaction or felt safety [41], others reported significant improvements in organisational culture and climate, as well as increased compassion satisfaction (being able to derive pleasure from your work) [42]. Nonetheless, this remains a core component of Substance Abuse and Mental Health Services Administration’s TIC principles [7]. It was encouraging to note how, in one project [32], staff support had been tailored to meet the needs of child welfare professionals by focusing on secondary traumatic stress and decision-making regarding the removal of children from biological parents.

In the domain of trauma-focused treatment/services, the introduction of trauma screening processes was relatively common. Although evaluation was limited and it was not always clear how, or if, this led to changes in the services provided, there was some evidence that this was perceived positively by professionals, resulting in substantial increases in the numbers of children screened at intake by child welfare and mental health professionals [31,32]. While this mainly positive evaluation mirrors the findings from trauma-informed screening/assessment initiatives in health services [43,44,45], implementation challenges were also noted with regard to IT systems, local culture, limited buy-in and the availability of evidence-based treatment. The importance of winning “hearts and minds” is evident in the wider TIC literature with some routine inquiry initiatives failing due to uncertainties about the benefits of screening, lack of clarity about how to use the information and respond to disclosures and concerns regarding a lack of availability of services for onward referral [46]. 

Importantly, a number of state-wide initiatives [25,29,31,32] specifically incorporated strategies to build capacity for assessment and trauma-focused treatment options alongside other implementation strategies, through training and dissemination of a range of evidence-based treatments. Likewise, the development of trauma-focused interventions outside of treatment was also evident with examples of a wide range of initiatives including trauma-informed assessment, group activities, psychoeducation, family-focused case management services and/or practical support services [20,21,23,24,32]. Two projects adapted the Attachment, Regulation and Competency (ARC) model [19,22], commonly used in residential facilities to build a therapeutic culture for the child within organisations [11], for use with adopted children and those involved with child protection services. Where evaluated, the provision of evidence-based treatment and/or trauma-focused services all demonstrated positive results for those who received them.

At the organisational level, leadership buy-in and strategic planning were the most commonly reported implementation components, although, as with other implementation domains, evaluation was often limited to qualitative findings. Despite these limitations, these elements of TIC implementation were highlighted as essential to success [25,32]. Top-down implementation models were more common and tended to favour the provision of initial training to agency directors and senior management, who then developed strategic implementation plans and structures [25,29,31,35]. However, one initiative, the Michigan Children’s Trauma Assessment Centre, emphasised a more “grassroots” approach, centred on developing community partnerships and implementation strategies based on extensive collaborative community assessments and consultation [32]. Changes to organisational/agency policy and procedures also featured in a number of initiatives, although only two projects provided specific examples of how policies, practice guides and processes were developed or revised to facilitate better alignment with TIC principles. 

Other components of organisational change were less prominent in the papers reviewed. Service user engagement formed part of only a few implementation strategies and primarily took the form of parent and carer inclusion in TIC training, although the Massachusetts Child Trauma Project also included service users in leadership teams [25] and the Michigan Children’s Trauma Assessment Centre involved parents and carers in initial community engagement and on-going consultation [32]. None of the 17 projects reported engaging with children and young people, nor making changes to offices or other facilities as a means of promoting a positive, safe physical environment in which to engage service users. This is in keeping with Bryson et al.’s [10] systematic review, which observed that involvement of service users seemed to occupy a less central role in the inpatient psychiatric and residential treatment TIC literature. Nonetheless, the review highlighted the effective and meaningful use of service user involvement through involving patients/service users in staff training [47,48,49]. Bryson et al.’s [10] review also noted a focus on making changes to the physical environment of treatment/residential spaces, to make them feel safe and welcoming for both patients/service users and staff. Changes made to the physical environment in one paediatric psychiatric hospital, which included repainting walls with warm colours, placement of decorative throws, rugs and plants and rearrangement of furniture to facilitate interaction [49], were found to be uniquely associated with a significant reduction in the rates of seclusion and restraint within the unit, suggesting that fairly minor and inexpensive changes can make a significant difference. It is interesting that this feature of TIC implementation was absent in the child welfare literature reviewed, pointing to an evident research gap. 

## 5. Conclusions

There is a robust and growing body of research which indicates that severe or chronic adversity in childhood has a significant, negative impact on a child’s development, health and life chances across their life course. Integrating awareness of childhood adversity into the public health agenda and cross-system service delivery (education, health, social care, justice) is therefore essential to prevent, recognise and mitigate the impact of trauma on children. This body of research also points to the importance of understanding parental/caregiver histories, who may also be impacted by early adversity. This knowledge has led to the development of models of trauma informed care in diverse practice settings in the USA which seek to mitigate the impact of adversity by promoting collaborative engagement with children and their caregivers and enhance child and family capacity for resilience and recovery. TIC has international reach with country-wide initiatives appearing across the UK and Europe. The rapid evidence review presented here sought, primarily, to explore the evidence pertaining to the organisational change processes required to implement trauma-informed care at a whole systems level within community-based child welfare settings. As part of a wider cross-system rapid evidence review of the trauma-informed implementation literature, twenty-one papers reporting on trauma-informed implementation at state/regional and organisational/agency levels of the child welfare system were identified. 

The diversity of interventions included in the papers reviewed and a variety of methodological difficulties, intrinsically limit the potential to draw firm conclusions from the literature. Notwithstanding these limitations, a number of multi-component evaluations of large-scale child welfare initiatives have begun to elucidate the positive impact of different TIC components. Training was the component of implementation most frequently evaluated with all studies reporting positive impact in terms of staff knowledge, skills and/or confidence. Other implementation components such as leadership and strategic planning, the development of evidence-based treatment and trauma-focused services, the provision of on-going staff support and the development of trauma-aligned policies and procedures, were also evidenced to varying degrees across many initiatives. The development of trauma-informed screening processes, and evidence-based treatments/trauma-focused services, where evaluated, all produced positive results. Staff support, service user engagement and changes to the physical environment were less prominent in the papers reviewed, pointing to evident research gaps. Whilst weaknesses in study design often limited generalisability, there was preliminary evidence for the efficacy of trauma-informed approaches in improving the mental and emotional well-being of children served by community-based child welfare services, as well as potential for reducing caregiver stress and improving placement stability. This body of literature, with good practice examples embedded within, will assist others in taking forward the important challenge of evidencing the effectiveness of TIC in child welfare settings. 

## Figures and Tables

**Figure 1 ijerph-16-02365-f001:**
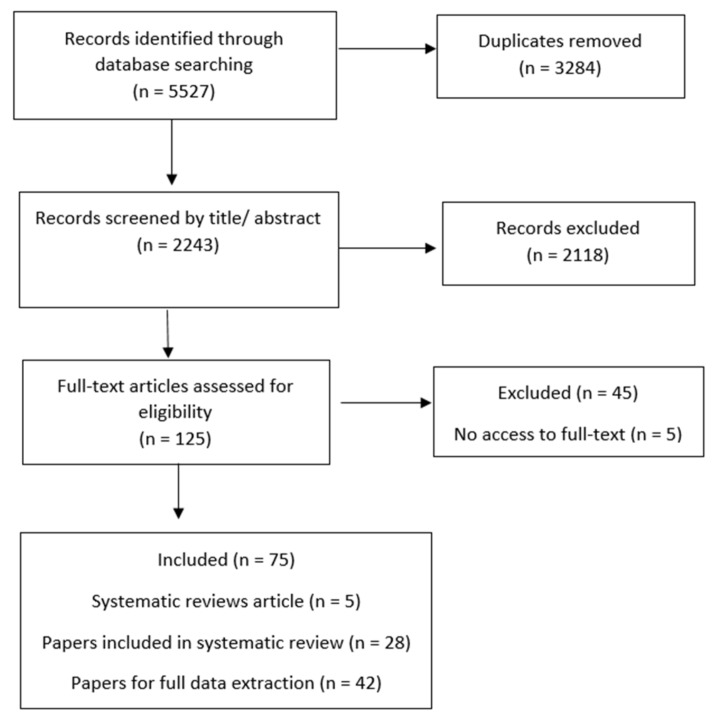
Search and extraction process.

**Table 1 ijerph-16-02365-t001:** Domains of trauma informed implementation in child welfare systems (Hanson and Lang, 2016) [9].

Workforce Development	Trauma-Focused Services	Organisational Change
• Training of all staff on the impact of abuse or trauma • Measuring staff knowledge/practice • Strategies/procedures to address/reduce traumatic stress among staff • Knowledge/skills in accessing evidence-based services	• Screening/assessment to identify trauma history and symptoms • Child’s trauma history included in case record/plan • Availability of evidence-based trauma-focused practices	• Collaboration, coordination, and information sharing (internal and external) • Procedures to reduce risk for client re-traumatisation • Promotion of consumer engagement • Provision of strength-based services • Safe physical environment • Written policies that include/support TIC principles

**Table 2 ijerph-16-02365-t002:** Implementation domains with evaluation data.

Project/Reference	Outcomes	Workforce Development			Trauma-Focused Services			Organisational Change:			
		Training	On-Going Support	Staff-Care	Screening/ Assessment	Evidence-Based Treatment	Trauma-Focused Practices/Services	Leadership Buy-In and Strategic Planning	Developing Policy, Procedures and Data Systems	Service User Involvement	Safe Physical Environment
Massachusetts Child Trauma Project [17,25,26]	x	x				x	x	x	x		
New Hampshire Adoption Preparation and Preservation/Partners for Change Project [18]	x						x				
Attachment, Regulation and Competency (ARC) Model [19]	x						x				
Indian Child Welfare Family Preservation Services [20]	x						x				
Project Kealahou (PK) [21]	x						x				
ADOPTS program [22]	x						x				
KVC Kansas [23,27]	x	x	x				x			x	
Intensive Permanence Services (IPS) [24]	x						x				
Creating Connections [28]		x									
Arkansas Initiative [29,30]		x									
The Connecticut Collaborative on Effective Practices for Trauma (CONCEPT) [31]		x		x							
Michigan Children’s Trauma Assessment Centre (CTAC) [32]					x	x		x	x		
Training for Adoption Competency (TAC) [33]		x									
Lemonade for Life [34]		x			x						
Sanctuary Model [35]		x		x							
Child Advocacy Centres Florida [36]		x									
Colorado, Connecticut, Massachusetts, Montana & North Carolina Trauma Screening Initiative [37]					x						

**Table 3 ijerph-16-02365-t003:** Empirical service user/treatment outcomes (*n* = 8).

Evaluation Design and Measures	Service User/Training Outcomes and Limitations
**Massachusetts Child Trauma Project** [17,26]	
**Design**: Preliminary implementation and follow up evaluations using a multi-source, multi-method approach. For service user outcomes this included: baseline assessment and six-month follow up of children referred to treatments (n = 326); and comparison of outcomes for 55,145 children who received the MCTP intervention (Cohort I; northern and western areas of the state) and 36,108 who did not (Cohort II; Boston and southern areas of the state) during the first year of implementation. **Measures**: child/family outcomes measured via administrative data on child maltreatment report, out-of-home placements, and adoption; clinical outcomes from children measured using the Posttraumatic Stress Disorder Reaction Index (UCLA PTSD-RI) and the Child Behaviour Checklist (CBCL).	**Outcomes**: • After approximately six months of EBT treatment, children had fewer post traumatic symptoms and behaviour problems. Children in the MCTP intervention group had fewer total substantiated reports of maltreatment, including less physical abuse and neglect than the comparison group by the end of the intervention year. • However, children in the MCTP intervention group had more maltreatment reports (substantiated or not) and total out-of-home placements than did their counterparts in the comparison group. • Assignment to MCTP was not associated with an increase in kinship care or adoption. **Limitations**: children were not randomly assigned to intervention; intervention and control children differ systematically in their background characteristics although this was accounted for by conducting an inverse probability of treatment weighted analysis.
**New Hampshire Adoption Preparation and Preservation/Partners for Change Project** [18]	
**Design**: Online and postal survey of licensed foster families, formerly licensed foster families and adoptive families from the past 10 years of records in one U.S. state (not specified). Aimed at examining whether foster and adoptive parent perceptions of the quality of trauma-informed child welfare and mental health services moderate the relationship between children’s behavioural health needs and parent satisfaction and commitment. Family units totalling 1206 were identified and 512 responded (42%: fostering only (n = 168), adoptive only (n = 215), fostering and having adopted (n = 66)). **Measures**: survey instrument designed by researchers.	**Outcomes**: • Trauma-informed mental health services (but not child welfare services) moderated the relationship between child behavioural health needs and foster parent (but not adoptive parent) satisfaction and commitment. • There was a significant interaction between child behavioural health needs and parent satisfaction and commitment (at low levels) of trauma-informed mental health services suggesting that these can buffer against low satisfaction and commitment, and thereby, potentially improve placement stability. **Limitations**: No standardised or validated measures. Based on adoptive parent and foster carer subjective perception of child behaviour problems and the quality of trauma-informed mental-health and child welfare services. Low response rate.
**Alaska Child Trauma Centre** [19]	
**Design**: naturalistic pre-test, post-test programme evaluation of treatment outcomes and placement stability in 93 children treated using ARC model (only 26 completed the intervention). **Measures**: Agency data and clinical assessments using Trauma Symptom Checklist Alternate Version, the UCLA PTSD Index for DSM IV and the Child Behaviour Checklist—CBCL used with all children. Administered at baseline, at three-month intervals, and at discharge.	**Outcomes** • The average drop in CBCL scores for children completing treatment was 19 points. • 90% children moved to permanent placements compared to usual 40%. **Limitations**: no specific comparison group so not clear how it compares to treatment as usual or if those completing treatment differed from those who did not, small numbers.
**Indian Child Welfare services** [20]	
**Design**: Evaluation of three years of family preservation services which served 73 families and 179 children over three years. Involved two projects (the RMQIC program and the SSUF program). **Measures**: Family functioning assessed via the North Carolina Family Assessment Scale (NCFAS), the Family Assessment Device, and the Parent Behaviour Inventory. Child safety measured directly by re-reports to CPS and indirectly through improvement on the Family Safety subscale of the NCFAS-AI (American Indian version of the NCFAS).	**Outcomes**: • A positive trend was seen in family safety for those families in the RMQIC program. • Families in the SSUF program showed significant positive change in the area of environment, and positive trends in the areas of caregiver capabilities, family safety and child well-being. • There were no re-reports during program services or within six months for any of the 49 families served by the RMQIC project. One new report within six months after services for the 24 families served by the SSUF project. This compared favourably with national re-report rates. • In the RMQIC project, 81% of families had their children maintained in the home, returned (if out-of-home-care was used), or placed with extended family members. • In the SSUF project, 96% of families were preserved with children either at home with parents (the most common result) or with extended family members. **Limitations**: no previous program baseline data presented and comparison only by national averages.
**Project Kealahou (PK)** [21]	
**Design**: Longitudinal design involving one to two hour-long structured interviews with youth and/or their caregivers at intake and at six-month intervals during the first two and a half years of PK services (September 2011–April 2014). Twenty-eight youth and 16 caregivers completed both baseline and six-month follow-up. **Measures**: Behavioural and Emotional Rating Scale, 2nd Edition (BERS–2C/2Y), Revised: Caregiver-Intake (CIQ-RC-I), Caregiver Strain Questionnaire (CGSQ), Child Behavior Checklist (CBCL 6–18), Columbia Impairment Scale (CIS), Education Questionnaire–Revision 2 (EQ–R2), Enrolment and Demographic Information Form (EDIF), Revised Children’s Manifest Anxiety Scale, Second Edition (RCMAS-2), Reynolds Adolescent Depression Scale, Second Edition (RADS–2), Youth Services Survey (YSS)	**Outcomes**: • Significant improvement from baseline to six-month follow-up on measures of youth strengths, competence, depression, impairment, behavioural problems, emotional problems, as well as caregiver strain. • A financial analysis indicated that these outcomes were obtained with a minimal overall increase in costs when compared to standard care alone ($365,803 vs. $344,141) **Limitations**: small number of participants, inability to determine which elements of PK services are responsible for its successful outcomes.
**ADOPTS program** [22]	
**Design**: Pre/post-test evaluation of the application of the ARC model with pre- or post-adoptive children and carers who had two or more lifetime traumatic exposures, with current post traumatic stress disorder (PTSD) and functional impairment in two domains. Twelve-month follow up. **Measures**: Clinician Administered PTSD Scale (CAPS); Trauma Symptom Checklist for Children (TSC-C); Behavioural Assessment System for Children (BASC); Parenting Stress Index (PSI).	**Outcomes**: • Significant lowering of Child Mental Health Symptoms with 76% of children assessed as having compared to 33.3% at follow-up. • The effect size for the reduction in PTSD symptoms was large (Cohen’s D = 1.88). • Significant reductions were found for child anxiety, depression, posttraumatic stress, dissociation and anger. • Significant reduction in care-giver stress. **Limitations**: lack of a control group, potential variability in treatment across clinicians, all evaluators were aware of treatment status of child.
**KVC Behavioural Healthcare Kansas** [23]	
**Design**: Longitudinal quasi-experimental study using administrative data to evaluate the impact of programme on 1499 children’s well-being and placement stability between over three of Trauma Systems Therapy (TST) implementations. **Measures**: KVC and researcher developed TST fidelity measures used to assess staff fidelity to TST implementation on a quarterly basis; child functioning was assessed by children’s caseworkers using the Child and Adolescent Functioning Assessment Scale (every 90 days), the Child Ecology Check-In (monthly basis); administrative placement history data were used to calculate children’s placement stability; fidelity scores and TST training dates of children’s care teams were used to calculate the level of TST or “dosage” that children received.	**Service User Outcomes**: • Increases in children’s exposure to TST (overall dosage) were associated with significantly greater improvements in functioning and behavioural regulation. • Increases in children’s exposure to TST (overall dosage) were not associated with greater improvements in emotional regulation; however, higher levels of fidelity to TST in children’s first quarter in KVC were associated with significantly greater improvements in emotional regulation. • In addition, TST fidelity in children’s first quarter in care, as well as increases in fidelity over time, were significantly associated with greater placement stability. • Increases across quarters in inner circle dosage (those who worked most closely with the children) were associated with significant improvements in children’s functioning and emotional regulation over time and increased placement stability. • Outer circle members’ implementation of TST in quarter one was significantly associated with improvements in functioning and placement stability. **Limitations**: Inability to randomly assign children to receive or not receive TST. The measure of TST dosage may not sensitively measure children’s level of exposure to TST. Reliance on secondary data to measure all outcomes.
**Intensive Permanence Services (IPS)** [24]	
**Design**: presented placement and relational outcome data from the initial pilot project in relation to 20 youth who had completed at least 13 months of the service. **Measures**: used discharge outcome data and the Youth Connections Scale (YCS) to measure relational permanence from time of service initiation and time of discharge.	**Service User Outcomes** • Of the young people who were involved in the pilot project and completed at least 13 months, 80% (n = 20) achieved legal permanency. Youth who were unable to complete IPS did not achieve legal permanency at this rate. Significant increase in scores on the Youth Connections Scale (YCS) from the time youth-initiated services to the time they were discharged. **Limitations**: small sample, no comparison group.

**Table 4 ijerph-16-02365-t004:** Empirical training outcomes (*n* = 8).

Evaluation Design and Measures	Training Outcomes and Limitations
**Massachusetts Child Trauma Project** [26]	
**Design**: Preliminary implementation and follow up evaluations using a multi-source, multi-method approach. For training, this included a single group pre-test/post-test training evaluation with 190 community mental health practitioners with one year follow up and 81% retention. **Measures**: training measured via Trauma-Informed System Change Instrument	**Outcomes**: • pre/post-test training evaluation found significant changes in perceptions of trauma-informed agency policy as well as perceptions of individual practices. **Limitations**: training outcomes self-reported only.
**Arkansas Initiative** [29,30]	
**Design**: Pre-test/multiple post-tests evaluation of training with child welfare leaders (n = 102, three month follow-up, retention 78%), with all child welfare staff (n = 438, follow up immediately after training, retention 93%) and a random sample of child welfare staff (n = 161, three month follow up, retention 88%). Additionally, half of the child welfare staff who were followed at three months were asked to complete a longer interview that asked about their success in implementing the action steps listed on their individualized plan developed at the end of training (n = 68). **Measures**: knowledge of trauma-informed practice and self-reported use of trauma-informed practices measured via questionnaire developed by authors.	Outcomes: • Significant increases in child welfare leaders’ knowledge about trauma-informed practice between pre-test and immediately post-test. • Significant increases in child welfare leaders’ self-reported use of trauma-informed practices between pre-test and three-month follow-up. • Child welfare staff’s knowledge of trauma-informed practice increased significantly between pre-test and post-test, as did self-reported changes in practice, although effect sizes were small when it came to direct support services for children and moderate for indirect support services. • 43.3% reported that they were able to fully implement trauma-informed strategies identified at training, while another 43.3% were partially implemented and 13.4% were unable to implement the strategy. **Limitations**: short follow-up period and outcomes based on self-reporting.
**The Connecticut Collaborative on Effective Practices for Trauma (CONCEPT)** [31]	
**Design**: pre-test/post-test with a stratified random sample of child welfare staff: 223 staff (45.2% response rate) completed the survey in Year 1 (pre-implementation) and 231 staff (46.5% response rate) completed the survey in Year 3. **Measures**: Perceptions of individual and agency capacity to provide trauma-informed care, measured via Trauma System Readiness Tool. Perceptions of individual and agency capacity to provide trauma-informed care.	**Outcomes**: • perceptions of individual and agency capacity to provide trauma-informed care increased significantly for 11 of the 12 domains. **Limitations**: response rate less than 50% for pre-test and post-test, based on self-reporting.
**Creating Connections** [28]	
**Design**: pre-test/multiple post-tests evaluation of training with staff conducting screening (n = 44, with follow-up immediately after training and at six months, retention 70.5%) and child welfare staff (n = 71, follow up immediately after training with child welfare). **Measures**: self-reported knowledge and skills gained via questionnaire developed by intervention developers.	**Outcomes**: • Screening staff knowledge and skills for administering the screening tools increased significantly and was retained at six-months follow-up. • Child welfare professionals self-reported competency scores on nearly all items, including the total item score, significantly improved from pre- to post-training **Limitations**: no longer-term follow up for child welfare staff training, all findings based on self-reporting.
**Child Advocacy Centres Florida** [36]	
**Design**: pre-test/multiple post-test design to evaluate training with staff who participated in training (n = 203, follow-up immediately post training and at one year, retention 12%). **Measures**: knowledge about trauma-informed care via questionnaire developed by intervention developers.	**Outcomes**: • knowledge about trauma-informed care increased significantly between pre- and immediately post-training and was retained after one year. **Limitations**: 12% pre-test/one-year post-test follow-up, poor retention rate.
**Lemonade for Life** [34]	
**Design**: Pre/post-test evaluation of pilot training with home visitors and parent educators in Kansas and Iowa (n = 17, follow up approximately six weeks after training completion, retention 71%). **Measures**: survey data, included items from the Hope Scale and Lemonade for Life-specific questions including: demographic information; participant experiences with ACEs personally and professionally; participant perceptions of using Adverse Childhood Experiences (ACEs) in work with families.	**Outcomes**: • Mean scores increased from pre to post in several areas: understanding how early experiences influence life course; home visitors’ knowledge of and self-reflection on their own ACEs score; and, knowing where to refer someone who is struggling with childhood adversity. The mean score on both Hope items (“I have the power to make my future better” and “I make others feel excited about the future”) decreased from pre to post—this was a new concept which may have led to a more realistic view evaluation of their own perspectives. **Limitations**: small sample, largely self-reported, only portions of the Hope Scale were used, no family outcomes.
**KVC Kansas** [27]	
**Design**: evaluation of fidelity to the Trauma Systems Therapy (TST) model following training based on training dates and fidelity scores of children’s care teams (caregivers, family service coordinators, caseworkers, supervisors and therapists) collected quarterly over three years of model implementation. **Measures**: researcher developed TST fidelity measures used to assess staff fidelity to TST implementation at quarterly basis; child functioning was assessed by children’s caseworkers using the Child and Adolescent Functioning Assessment Scale (every 90 days), the Child Ecology Check-In (monthly basis); administrative placement history data were used to calculate children’s placement stability; fidelity scores and TST training dates of children’s care teams were used to calculate the level of TST or “dosage” that children received.	**Outcomes**: • 384 KVC staff members were trained during the course of the first formal trainings and approximately 69% of KVC’s 397 foster parents over the course of the study period. • Average TST “dosage” scores for each member of children’s care-teams indicate that on average from 2012 to 2014 KVC staff implemented TST with increasing fidelity, with the average dosage score for children’s care teams steadily increasing from 7.95 (SD = 2.25; out of 30) at the start of the roll-out (first quarter of 2012) to 20.77 (SD = 5.67) at the last quarter of 2014. **Limitations**: based on self-report.
**Training for Adoption Competency (TAC)** [33]	
**Design**: Evaluation of training fidelity using observation and feedback and pre/post-test evaluation of training outcome which involved 855 participants including mental health professionals, public and private mental health agencies, adoption-specialty organizations, family service agencies, private practices, child welfare agencies residential treatment facilities and other settings. Training outcomes assessed mid training and end of training—timing not specified. Reference to control group but details not provided. **Measures**: training outcomes measured/assessed via mid training and end of training surveys of participants and a self-assessment of adoption competency administered at the conclusion of the modules as a retrospective pre- and post-assessment.; training fidelity assessed using fidelity observations and feedback from participants and trainers to assess the quality and relevance of training and the fidelity of curriculum delivery.	**Outcomes**: • More than 300 fidelity observations of training delivery across 59 cohorts confirm full delivery, with fidelity, of nearly 100% of all content of all modules • TAC participants experienced an average gain in pre- to post-test scores of 46.08 points, while those in the control groups of comparably qualified professionals experienced a gain of only 1.58 points. • There was not a statistically significant difference in test scores between participant and control groups at pre-test. • There was a significant interaction between the training and time on test scores. • Based on 1148 responses containing 4928 separate narrative descriptions of the ways practices were influenced by the training, all TAC participants reported change in at least two of the six defined aspects of practice; 59.88% reported change in all five aspects at the individual clinician level and 51.75% reported that TAC influenced the procedures, programming and/or services in their organization. **Limitations**: measures of training outcomes and changes in practice primarily self-report, details of controls not provided and sample size not always clear.

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
