# Peer review of "Trauma Informed Child Welfare Systems—A Rapid Evidence Review"

_ijerph, 2019, doi:10.3390/ijerph16132365_

Round 1

Reviewer 1 Report

This is a well written paper. The paper contains new and significant information adequate to justify publication. The paper demonstrates an adequate understanding of the relevant literature in the field and cite an appropriate range of literature sources. The results are presented clearly and analyzed appropriately.

Some suggestions:

-Please write the title of Table 1 in line 96, because it is confuse, considering that there is another table 1, in the supplementary material.

-There are some references missing? Please see lines 253, 270, 351, 360, 365, 370, and so on, the brackets are empty.

- Line 423 child is written twice. 

Author Response

Thank you for your comments - they have been addressed as follows (see bold)

- Please write the title of Table 1 in line 96, because it is confuse, considering that there is another table 1, in the supplementary material - inserted

-There are some references missing? Please see lines 253, 270, 351, 360, 365, 370, and so on, the brackets are empty - references have been revised throughout

- Line 423 child is written twice - this has been amended

Thank you 

Lisa

Reviewer 2 Report

Given the enormous interest in trauma informed care, and the increasing numbers of publications, this manuscript will be useful to the field.  That said, I have a few comments that I hope you will find helpful.

Overall, the manuscript needs thorough proof reading and editing.  There were numerous places with mistaken words and incorrect punctuation.  One example was some inconsistency in your use of a hyphen in trauma informed care.  As a rule, I generally don't hyphenate the noun (trauma informed care) but hyphenate the adjective (trauma-informed...).  Consistency is obviously the key. Please spell out the acronym KVC the first time (line 236).

In terms of content and flow, the organization of the material was fine.  I liked your use of subheadings.  There are places where the writing needs to be tightened.  In fact, there are many sentences that are so long that they become awkward and or confusing (see the first sentence in the section on training 3.2.1 as an example).  Please review long sentences to determine if they can be made shorter and more clear.

It would also be helpful if you could put some of the findings in a table.  I recognize that the outcomes you are reporting vary, but it became difficult to track the overall results.  Perhaps your overall findings could be summarized in a table.

Author Response

Thank you for your comments - these have been addressed as follows (see bold):

Overall, the manuscript needs thorough proof reading and editing.  There were numerous places with mistaken words and incorrect punctuation.  One example was some inconsistency in your use of a hyphen in trauma informed care.  As a rule, I generally don't hyphenate the noun (trauma informed care) but hyphenate the adjective (trauma-informed...).  Consistency is obviously the key. The paper has been thoroughly proofed and edited by colleagues (see tracked changes on manuscript). 

Please spell out the acronym KVC the first time (line 236). - KVC  is not an acronym but the name of the institution - its has been amend to "KVC Behavioral Healthcare" for clarity. 

In terms of content and flow, the organization of the material was fine.  I liked your use of subheadings.  There are places where the writing needs to be tightened.  In fact, there are many sentences that are so long that they become awkward and or confusing (see the first sentence in the section on training 3.2.1 as an example).  Please review long sentences to determine if they can be made shorter and more clear.The paper has been thoroughly proofed and edited by colleagues you has amended overly long sentences throughout (see tracked changes on manuscript). 

It would also be helpful if you could put some of the findings in a table.  I recognize that the outcomes you are reporting vary, but it became difficult to track the overall results.  Perhaps your overall findings could be summarized in a table - i have inserted 2 tables with empirical findings related to service user/treatment outcomes (table 3) and training outcomes (table 4). The remaining findings are mainly descriptive and are discussed in narrative form. My only concern is that this makes the paper overly long - there often multiple methods/outcomes which need to be described - I am happy to be guided by you on this. 

thanks

Lisa